# Cannabidiol and Beta-Caryophyllene in Combination: A Therapeutic Functional Interaction

**DOI:** 10.3390/ijms232415470

**Published:** 2022-12-07

**Authors:** Henry Blanton, Linda Yin, Joshua Duong, Khalid Benamar

**Affiliations:** 1Department of Pharmacology and Neuroscience, School of Medicine Lubbock, Texas Tech University Health Sciences Center, Lubbock, TX 79430, USA; 2Garrison Institute on Aging, Texas Tech University Health Sciences Center, Lubbock, TX 79430, USA

**Keywords:** cytokines, inflammation, CBD, BCP, synergy, pain, cannabis, terpenes, cannabinoid

## Abstract

Cannabis contains over 500 distinct compounds, which include cannabinoids, terpenoids, and flavonoids. However, very few of these compounds have been studied for their beneficial effects. There is an emerging concept that the constituents of the cannabis plant may work in concert to achieve better therapeutic benefits. This study is aimed at determining if the combination of a minor cannabinoid (cannabidiol, CBD) and a terpene (beta-caryophyllene, BCP) works in concert and if this has any therapeutic value. We used an inflammatory pain model (formalin) in mice to test for any functionality of CBD and BCP in combination. First, we determined the analgesic effect of CBD and BCP individually by establishing dose-response studies. Second, we tested the analgesic effect of fixed-ratio combinations and monitored any adverse effects. Finally, we determined the effect of this combination on inflammation. The combination of CBD and BCP produces a synergistic analgesic effect. This effect was without the cannabinoid receptor-1 side effects. The analgesic effect of CBD and BCP in combination involves an inflammatory mechanism. The combination of these two constituents of the cannabis plant, CBD and BCP, works in concert to produce a therapeutic effect with safety profiles through an inflammatory mechanism.

## 1. Introduction

Cannabis contains over 500 distinct compounds, which include cannabinoids, terpenoids, and flavonoids [1]. However, very few of these chemicals have been studied for their beneficial effects alone or in combination.

Although cannabidiol (CBD) is an isomer of Δ^9^-tetrahydrocannabinol (Δ^9^-THC), the primary psychoactive component in *Cannabis sativa*, CBD has been reported to produce many medicinal benefits, without the intoxicating effect characteristic of Δ^9^-THC [2]. The pharmacology of cannabidiol is complex and remains under investigation [3]. Notably, CBD displays weak affinity for the cannabinoid receptor-1 (CB_1)_ and cannabinoid receptor-2 (CB_2_) [4], and instead is thought to produce effects through a various other targets. Mounting preclinical evidence reveals that CBD has analgesic effects [5,6,7].

Despite the wide adoption of CBD for pain, clinical trials to date have failed to demonstrate significant analgesic effects with pure CBD [8,9]. However, it has been suggested that compounds in the cannabis plants function more efficiently in concert with each other, rather than in isolation [10].

In cannabis, CBD is only one chemical constituent out of hundreds of active or potentially active compounds including other cannabinoids, as well as terpenes, the compounds responsible for the aroma of cannabis [11]. In addition to contributing, to the aroma of cannabis, many of the terpenes found in cannabis are bioactive compounds. One of the most abundant terpenes in cannabis is beta-caryophyllene (BCP), a sesquiterpene that is also found in black pepper and clove [11]. BCP was determined to be a natural selective CB_2_ receptor agonist [12]. It has several beneficial effects such as analgesia and anti-inflammation [10,12,13,14,15,16,17]. 

The rapidly shifting policies affecting access to cannabinoids and interest in the possible medicinal use of cannabis (particularly for pain management) create an urgent need to assess the beneficial and adverse effects of the therapeutic use of cannabinoids. In particular, the combination of compounds functions more efficiently in concert with each other rather than in isolation. Indeed, although limited, there is evidence of an interaction of cannabinoids found in the cannabis plant.

For example, CBD and Δ^9^-THC work synergistically in chemotherapy [18] and sciatic nerve injury pain models [5].

However, there is also the unexplored possibility that other compounds (non-cannabinoid) may also work in concert with cannabinoid to produce more efficacious therapy with safety profiles. 

This study is aimed at determining if the combination of a minor cannabinoid (CBD) and a terpene (BCP) works in concert and if this has any therapeutic value.

## 2. Results

### 2.1. CBD and BCP in Combination Produced Synergistic Analgesic Effect

The design of the experiment is shown schematically in Figure 1A. CBD was administered i.p. at doses of 1, 2.5, 10, 25, and 50 mg/kg (*n* = 6). Administration of CBD (Figure 1B) resulted in dose-dependent reductions in pain behaviors in the inflammatory phase of the formalin test from minutes 20 to 40 post-formalin injection. The ED_50_ value for CBD was 2 mg/kg and the Hill slope-0.9.

The BCP was administered at doses of 1, 3, 10, and 30 mg/kg. Administration of BCP (Figure 1C, *n* = 6) resulted in dose-dependent reductions in pain behaviors in the inflammatory phase of the formalin test from minutes 20 to 40 post-formalin injection. The ED_50_ value was 2.2 mg/kg for BCP and the Hill slope-0.88. Various combinations of the two compounds were tested at this fixed-dose ratio (based on the ED_50_ of each compound) (Figure 1E, *n* = 8). The combination produced dose-response suppression of formalin-induced pain. The ED_50_ for the CBD:BCP combination was calculated to be 1.4 mg/kg (Figure 1D), which was lower than the ED_50_ of the individual effect. These data demonstrate that the combination is more potent than the individual effect. Of great interest is that the combination of a lower dose of CBD (2 mg/kg) and CBD (2.2 mg/kg) (Figure 1E, 4.4 ± 0.4) produced a superior maximal effect than that achieved by a high dose of the individual effect of CBD (10 mg/kg, 6.2 ± 0.8, Figure 1C) and BCP (30 mg/kg, 6.2 ± 0.7, Figure 1D). In addition, isobol analysis showed that this combination is synergistic (Figure 1F).

We presented the data from the second phase of the formalin test (from minutes 20 to 40 post-formalin injection) because no effect was observed in the first phase (0–15 min post-formalin).

We evaluated the effects of BCP and CBD alone and in combination to determine if the compounds, individually or in combination, affected body temperature, locomotor activity, or motor coordination. One-way ANOVA showed no significant effects caused by drug treatment that were seen in the body temperature (F_3,20_ = 0.5593; *p* = 0.6480) (Figure 1G), open field test (F_3,20_ = 0.2060; *p* = 0.8910) (Figure 1H), or rotarod test (F_3,20_ = 0.0773; *p* = 0.9715) (Figure 1I) compared to vehicle.

### 2.2. Effects of BCP, CBD, and Combination on Formalin-Induced Cytokine Expression in Plasma

Cytokine expression following intraplantar injection of formalin was analyzed in mice treated with vehicle, BCP (2.2 mg/kg), CBD (2 mg/kg), or the combination of BCP (2.2 mg/kg) + CBD (2 mg/kg). Two-way ANOVA revealed a significant effect of drug treatment on cytokine expression (Figure 2, F_3,204_ = 17, *p* < 0.0001, *n* = 4). CBD (2 mg/kg) reduced expression of seven cytokines (Figure 2A): IGFBP-3 (*p* = 0.0407), LIX (*p* = 0.0037), Pentraxin 2 (*p* = 0.0004), PCSK9 (*p* = 0.0472), REG3G (*p* = 0.0001), E-selectin (*p* = 0.0090), and VEGF (*p* = 0.0166). 

BCP (2.2 mg/kg) reduced expression of five cytokines (Figure 2): C1qR1 (*p* < 0.0001), IGFBP-3 (*p* = 0.0001), Pentraxin 2 (*p* = 0.0001), REG3G (*p* = 0.0070), VEGF (*p* = 0.0004). 

The CBD (2 mg/kg) + BCP (2.2 mg/kg) combination reduced expression of 16 cytokines (Figure 1A): EGF (*p* = 0.0349), ICAM 1 (*p* = 0.0371), IGFBP-1 (*p* = 0.0097), IGFBP-3 (*p* = 0.0020), IGFBP-5 (*p* = 0.0078), IGFBP-6 (*p* = 0.0002), IL-33 (*p* = 0.0090), LIX (*p* = 0.0001), MMP-2 (*p* = 0.0018), MMP-3 (*p* = 0.0001), Pentraxin 2 (*p* < 0.0001), PCSK9 (*p* < 0.0001), REG3G (*p* < 0.0001), E-selectin (*p* = 0.0059), VCAM-1 (*p* = 0.0442), and VEGF (*p* < 0.0001).

The list of cytokines (positions of 111 cytokines and positive and negative controls in duplicate) tested is presented in Figure 3A. Colored dots indicate a statistically significant decrease in drug treatment. Representative array blot is shown in Figure 3B.

## 3. Discussion

In light of our research, we provide the first scientific evidence supporting that the combination of these two constituents of the cannabis plant, CBD and BCP, works in concert to produce a therapeutic effect with safety profiles through an inflammatory mechanism.

We tested fixed ratio combinations (based on the ED_50_ of each compound) and, using isobolographic analysis, we found that the analgesic effect of CBD and BCP in combination is synergistic. These data indicate that not only can cannabinoids found in cannabis plant work in concert, but such an effect extends to other components of this plant as terpenes. The ED_50_ for CBD and BCP in combination is smaller than the individual effect, indicating that this combination is more potent than CBD and BCP alone. These data suggest that this combination has the potential to serve as an alternative analgesic therapy.

We also tested this combination for the CB_1_-associated side effects. We found that CBD and BCP in combination did not produce hypothermia, sedation, or motor incoordination. Thus, administering CBD in combination with BCP is devoid of CB_1_-associated side effects. The other advantages of this combination are (1) the anti-depressive effects of CBD [19] and BCP [20,21,22] are of additional benefit for chronic pain conditions (because depression is known to be a feature of chronic pain in humans); and (2) neither BCP nor CBD is a controlled substance, which could lead to rapid translation of this combination to patients. 

The mechanisms underlying this synergistic interaction remain to be determined. Proteomic analysis shows that the enhanced analgesic effect of the combination involved an anti-inflammatory mechanism. The number of inflammatory mediators affected by the combination was 17 and was superior in comparison with the individual effect of CDB (7) and BCP (5). Of great interest is that some inflammatory mediators are uniquely affected by the combination (e.g., EGF, ICAM-1, IGFBP-1, IGFBP-5, IGFBP-6, MMP-2, and MMP-3, VCAM1) compared to the individual effect of CBD and BCP. These data support the action of this combination on these inflammatory mediators as a potential mechanism by which CBD and BCP work in synergy.

We focused on the inflammation because an anti-inflammatory component of BCP and CBD individually has been reported [10,23,24,25,26], but this does not negate the role of other mechanisms. Various potential mechanisms may contribute to the synergistic interaction between CBD and BCP. The transient receptor potential vanilloid 1 (TRPV1) ligand-gated ion channel [27] and the serotonin 1A receptor (5-HT_1A_) G-protein-coupled serotonin receptor are targets of CBD [28]. BCP is a selective CB_2_ receptor agonist [12]. A potential cross-talk between these GPCRs and, consequently, enhancement of the down-stream signaling may play a role in the synergistic analgesic effect of the combination. Pharmacokinetics interaction between CBD and BCP could also play a role. 

In summary, the present data support the concept that cannabinoids and terpenes work synergistically with therapeutical value. In particular, the CBD and BCP in combination produce a synergistic analgesic effect without CB_1_-associated side effects. Finally, scientific investigation of the interactive effects of CBD and BCP on a therapeutic target has not been undertaken. Therefore, our present data set the stage for future studies on the therapeutic value of this combination in other diseases and testing other terpenes in combination with other cannabinoids or terpenes.

## 4. Material and Methods

### 4.1. Animals

Experiments were performed using 16-week-old male C57BL6/J mice purchased from Jackson Laboratories (Bar Harbor, ME, USA). Mice used in these experiments were group-housed (four per cage) under a 12:12 h light–dark cycle (lights on 07:00, lights off 19:00) and provided with standard mouse chow ad libitum. All animal care and experimental procedures used in this study were approved by the Institutional Animal Care and Use Committee (IACUC) of the Texas Tech University Health Sciences Center and conducted in accordance with the National Institutes of Health accepted guidelines found in the Guide for the Care and Use of Laboratory Animals.

We used 244 mice in this study (Table 1).

### 4.2. Drugs

CBD was purchased from Cayman Chemical (Ann Arbor, MI, USA). BCP was purchased from Sigma-Aldrich (St. Louis, MO, USA). BCP and CBD were dissolved in a vehicle consisting of 5% DMSO, 5% ethanol, 5% Tween80, and 85% saline. Drugs were administered via intraperitoneal (i.p.) injection 30 min prior to testing.

### 4.3. Formalin Test

Subcutaneous injection of 10 µL of 2.5% formalin was injected into the hind paw of the mouse, and pain behaviors were recorded over the course of one hour. Pain behaviors for the injected paw were divided into three categories: (1) favoring (little or no weight-bearing); (2) lifting (paw has no contact with any surface); and (3) licking, biting, shaking, or rapid lifts of the paw. For each five-minute (300 s) interval, a composite pain score was calculated by analyzing the amount of time (seconds) spent displaying these behaviors using the composite pain score-weighted scores technique [29]. The composite pain score for each 5 min interval was calculated using the formula: CPS = ((2 × Behavior 3) + (Behavior 2)/300).

For the inflammatory phase of the formalin test, the area under the curve (AUC) was calculated for the composite pain score values between minutes 20 to 40 following formalin injection. 

All the experiments were done blinded. The investigator who conducted the pain-like behavior was unaware of allocation groups to ensure that all animals in the experiment were handled, monitored, and treated in the same way.

### 4.4. Isobologram

An isobologram was derived by plotting an XY graph with the ED_50_ for CBD on the Y axis, the ED_50_ for BCP on the X axis, and connecting the two points diagonally, through the so-called “line of additivity” [30]. If drug synergy is displayed, the ED_50_ for the drug combination falls below the line of additivity, indicating drug synergy. 

### 4.5. Tetrad Behaviors

A cohort of mice was used to evaluate the potential central nervous system Δ^9^-THC-like side effects of CBD, BCP, and their combination. Mice were evaluated for drug-induced changes in body temperature, locomotor activity, and motor function. 

#### 4.5.1. Open Field Test

Mice were placed into a corner of a rectangular open field test arena (40 cm length, 40 cm width, 30 cm height). A video camera was mounted on a tripod positioned above the center of the arena, and the mice were allowed to explore the arena for 5 min. The total distance traveled in centimeters during the 5 min exploration period was tracked from each video file and calculated using EthovisionXT video-tracking software (Noldus Information Technology, Wageningen, The Netherlands). 

#### 4.5.2. Body Temperature

Body temperature was evaluated using a rectal thermometer (Physitemp; Clifton, NJ, USA).

#### 4.5.3. Rotarod

Motor impairment was tested using the accelerating rotarod test (Model LE8205, Harvard Apparatus, Holliston, MA, USA). Mice were placed on a rotating rod revolving at 4 RPM, which was programmed to accelerate to 40 RPM over the course of 5 min. The latency to fall from the rotarod was recorded in seconds. Mice were trained for three days on the rotarod before experimental testing began. 

### 4.6. Cytokine Analysis

Mice were sacrificed 35 min after formalin injection and blood was collected in 2 mL microcentrifuge tubes. Analysis of cytokines from plasma was performed using a Mouse XL Cytokine Array Kit (R&D Systems; Minneapolis, MN, USA). 

This kit allows simultaneous measurement of 111 mediators/markers in a single sample. The blood was allowed to clot at room temperature for 2 h prior to being centrifuged at 15,000 RPM at 4 °C for 15 min. Following centrifugation, the supernatant serum was collected and transferred to a new 2 mL microcentrifuge tube. The cytokine arrays were prepared according to the instructions provided by the manufacturer; 130 µL of serum was used for each array. The arrays were visualized using a digital imaging system (Azure Biosystems C400, Dublin, CA, USA). A digital image of each array was captured and analyzed using ImageJ image analysis software (National Institutes of Health; Bethesda, MD, USA). For each array, the relative expression of each cytokine was determined by analyzing the average gray value (0–255) for each spot on the array, beginning with the positive (reference) spots and negative (blank) spots. The average gray value for the positive and negative controls was determined. Next, for each spot on the array, relative signal intensity was determined using the formula:Relative intensity = ((measured gray value − average gray value of negative control)/average gray value of positive control))/100.

### 4.7. Statistics

Statistics were performed in Graph Pad Prism 9 (GraphPad Software; San Diego, CA, USA). Two- and one-way ANOVA with Dunnett’s post hoc tests. Significance was set at *p* < 0.05. 

## Figures and Tables

**Figure 1 ijms-23-15470-f001:**
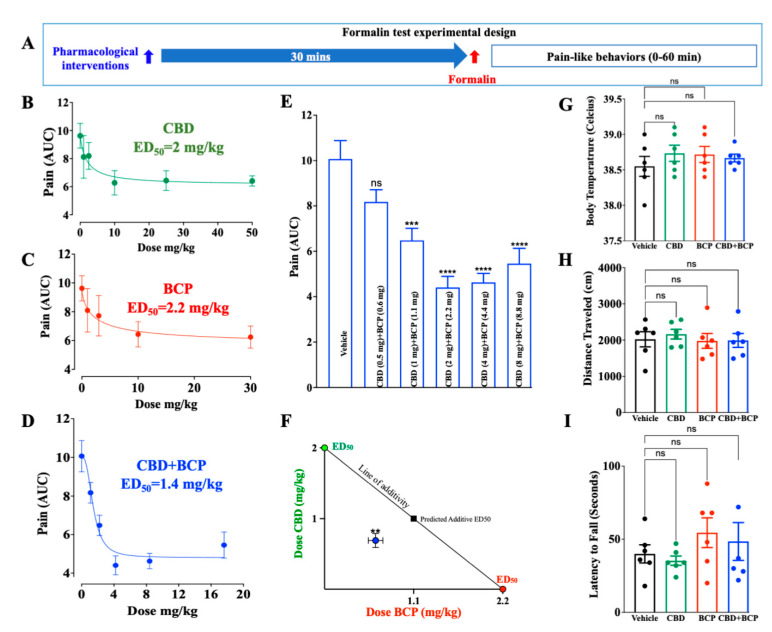
Experimental design (**A**). CBD (**B**) and BCP (**C**) individually and combined (**D**,**E**) produce dose-dependent analgesia. Isobolographic analysis of the CBD and BCP in combination (**F**). Effects of BCP and CBD alone and in combination on body temperature (**G**), locomotor activity (**H**), and motor coordination (**I**), *n* = *** *p* < 0.001, **** *p* < 0.0001 (compared to vehicle), ns: not significant (compared to vehicle).

**Figure 2 ijms-23-15470-f002:**
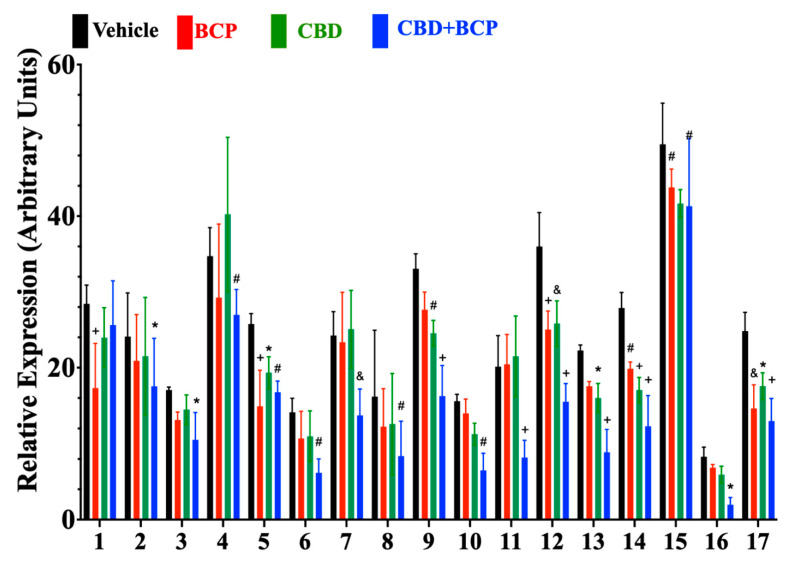
Effects of BCP, CBD, and combination on cytokine expression (*n* = 4). List of cytokines 1 (C1qR1), 2 (EGF), 3 (ICAM-1), 4 (IGFBP-1), 5 (IGFBP-3), 6 (IGFBP-5), 7 (IGFBP-6), 8 (IL-33), 9 (LIX), 10 (MMP-2), 11 (MMP-3), 12 (pentraxin 2), 13 (PCSK 9), 14 (REg3G), 15 (E-selectin), 16 (VCAM-1), and 17 (VEGF). Bars correspond to the group means ± SEM. * *p* < 0.05; # *p* < 0.01; & *p* < 0.001; + *p* < 0.0001.

**Figure 3 ijms-23-15470-f003:**
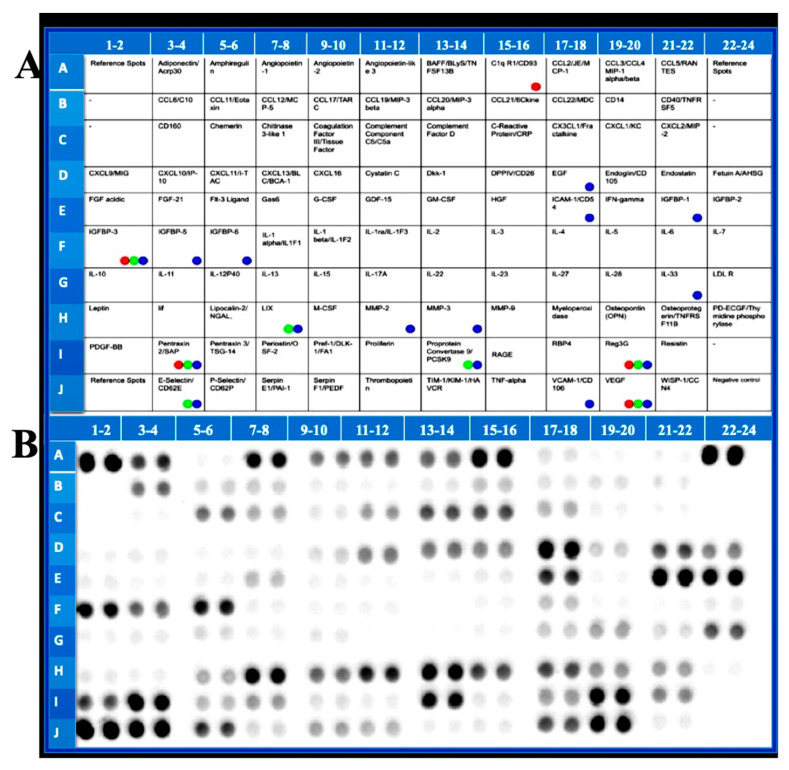
The list of cytokines measured by cytokine array (**A**). Colored dots indicate a statistically significant decrease by drug treatment. Representative array blot (**B**).

**Table 1 ijms-23-15470-t001:** Groups, treatment and dose.

Experiment	Treatment	Dose	Mice/Group	Total Mice Number
**Formalin**	CBD	1–50 mg/kg, i.p.	6	42 (6 × 6 + 6 vehicle)
	BCP	1–30 mg/kg, i.p.	6	42 (6 × 6 + 6 vehicle)
	Combination	1.1–16.8 mg/kg, i.p.	8	56 (6 × 8 + 8 vehicle
**Tetrad**	CBD	2 mg/kg, i.p.	4	16 (4 × 4)
	BCP	2.2 mg/kg, i.p.	4	16 (4 × 4)
	Combination	4.2 mg/kg, i.p.	4	16 (4 × 4)
				Vehicle group 4
**Cytokines**	CBD	2 mg/kg, i.p.	4	16 (4 × 4)
	BCP	2.2 mg/kg, i.p.	4	16 (4 × 4)
	Combination	4.2 mg/kg, i.p.	4	16 (4 × 4)
				Vehicle group 4
				**Total = 244**

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
