# Peer review of "Cannabidiol and Beta-Caryophyllene in Combination: A Therapeutic Functional Interaction"

_ijms, 2022, doi:10.3390/ijms232415470_

Round 1
Reviewer 1 Report
Introduction:
1. CBD in fact behaves as an iverse agonist toards CB2 receptors. Please correct the sentence.
2. It would be good if the Authors could present in details CBD in terms of its structure and localization.
Methodology:
1. Please provide the total number of animals used in the study.
2. Unfortunatley, there is no information regarding the experimental groups (number, types in terms of treatment, drug concentration, etc.). This should be placed in the methodology section rather than in the Results.
3. In line with this, please provide the time point at which the drug and its combination were injected.
4. From the texts it is assumed that the Authors observed the effect during the late phase of inflammation. What about the acute phase?
5. Were there use additional group of animals for the study of tetrad behaviors. If yes, please provide the total number of animals, group division etc. If no, then please state whether the drugs and the combination (?) were administered again. Otherwise, both the rotarod and open-field take time and the Authors did not provide information on the time for which the drug(s) is active (no enzymatic degradation, etc.). Obviously, there were no possibilities to performed each study (formalin test, open-field etc.) simultaneously.
6. Subsection 2.6. The Authors examined the level of cytokines. However, was it done for the formalin-induced animals?
7. What was the source of the blood?
8. In line with the bove-mentioned, if the Authors collected blood from FA-induced inflammatory pain animals, and between the collection also other tests were performed (i.e., tetrad), then what was the reason to examine the cytokines? It is known that the level of both pro- and antiinflammatory cytokines rapidly changed duriing the inflammation, and the first phase (acute one) in crucial do observe any changes and how the body/drug response.
Results:
1. The 3.1 subsection is written in a rather chaotic manner. Please check "The ED50 value was 2 mg/kg for BCP was administered at doses of 1, 3,
10, and 30 mg/kg. Administration of BCP (Figure 1C) resulted in dose-dependent reductions in pain behaviors in the inflammatory phase of the formalin test from minutes 20 to 40 post-formalin injection. The ED50 value was 2.2 mg/kg for BCP".
Reviewer 2 Report
This is an interesting study of the interaction between cannabis constituents in an acute model of inflammatory pain. The use of the non-psychoactive constituent CBD and a terpene is particularly interesting. The data is well described and largely supports the study claims, however, only has limited amount of data and is a really a brief report.
There are only minor comments
- Abstract – ‘several fixed-ratio combination’. The study only examined one fixed-ratio combination in the isobolographic component
- Abstract and results. It is claimed that there was a synergistic analgesic interaction between CBD and BCP. However, there is no statistical comparison between the theoretical ED50 and the experimentally obtained ED50 for the the fixed ratio of CBD and BCP.
- Results. Lines 143-7, 158-160 – these sentence should be in the methods
- Results- what is the second paragraph about ‘We presented … post-formalin).’
- Results – stats are needed to justify ‘In addition … synergistic (section 2.4 and Fig 1F).’ This is particularly important given the error bars for the ED50 of the combination
- Results – don’t need to list all of the doses (155-157), it is enough to state the fixed ratio.
- Results – the authors have used linear isobolographic analysis. However, Talarida’s work states that this can only be used if the two drugs to have a similar. This is clearly not the case in Fig 1B-D. If the authors cannot use non-linear analysis they should at least identify it as a potential issue as it can drastically altered the theoretical ED50 of the CBD/BCP combination.
- The discussion is too brief and inadequate. The authors should expand the discussion to include other studies which have described analgesic synergy between cannabis constituents. The authors should also add a paragraph discussing potential mechanisms based on their findings and prior studies, eg how are the inflammation mechanisms related to the pain scores etc etc.
Minor comments
- check all figures. Fig 1F is incorrectly labelled. Fig 2B is impossible to read due to the small text. Fig 2A id difficult to follow – could it be formatted in another way
Reviewer 3 Report
I would like to thank the Authors and the Editors for the opportunity to comment on this very interesting paper, in which combinations of two Cannabis-derived compounds, CBD (a non-psychotropic cannabinoid) and BCP, of analgesic and anti-inflammatory effect, have been tested on lab mice by registering behavioral outputs and chemokine induction in a 60 minute interval after formalin injection.
The manuscript is very interesting and of relevance for future pharmacological interventions in pain management and possibly depressive symptoms, revealing that the two compounds display a synergistic effect and their combined dosage is less than the amount required if only one was used. The text is clear in all its parts; the experimental conditions seem adequate and the statistical analysis is sound. I have no major comments in this regard. I would just note that:
- lines 57-58: this sentence is incomplete and makes no sense
- lines 135-136: there are specifications about putting formulas in the text
- Figure 1: there are two panels indicated by the letter "E"
- Figure 2: panels B and C may be part of a separate Supplementary Figure, as they are unreadable in the paper; also, a Table could be used to summarize the results presented in paragraph 3.2, to make for a better read.
- as a suggestion, I would propose that the Discussion paragraph could be slightly expanded by putting these discoveries in the context of other experimental/genetic/psycho-behavioural work on both mice and humans and on the other effects of CBD, for example sleep/insomnia, PTSD and anxiety management.
Round 2
Reviewer 1 Report
1. The table 1 should be placed in the manuscript. Unfortunately, I can't find it in the text.
2. Total number of animals should also be provided in the methodology (animals)
3. In order to determine any reliable reaction in terms of cytokines level, at least 3 time points should be taken. Immediately after drug administration, at about 5 min pos-injection and i.e. 35 as stated in the text.
Otherwise, these results seem to be insignificant.
4. Also, the Authors should provide information in the text that in the acute inflammatory phase there were no significant differences.
Author Response
We appreciate the reviewers' thorough review, helpful suggestions, and constructive criticism on the manuscript.
- Table 1 should be placed in the manuscript. Unfortunately, I can't find it in the text.
Table 1 is now included in the method section
- Total number of animals should also be provided in the methodology (animals)
This information is now included in the Methos section.
- In order to determine any reliable reaction in terms of cytokines level, at least 3-time points should be taken. Immediately after drug administration, at about 5 min pos-injection and i.e. 35 as stated in the text. Otherwise, these results seem to be insignificant.
The formalin test consists of acute pain phase 1 (minutes 0–15, non-inflammatory) and inflammatory pain phase 2 (minutes 15–60). The inflammatory response to formalin injection starts at 20 mins post-CFA. We agree with the reviewer about the need for other time points. However, given that no inflammation occurs during the acute phase (minutes 0–15, non-inflammatory), we believe there is no need to analyze cytokine levels at 5 mins post-formalin.
- Also, the Authors should provide information in the text that in the acute inflammatory phase, there were no significant differences.
Please see the result section, paragraph 3